# Reservoir Boosting : Between Online and Offline Ensemble Learning

**Leonidas Lefakis**
Idiap Research Institute
Martigny, Switzerland
leonidas.lefakis@idiap.ch

**François Fleuret**
Idiap Research Institute
Martigny, Switzerland
francois.fleuret@idiap.ch

## Abstract

We propose to train an ensemble with the help of a reservoir in which the learning algorithm can store a limited number of samples.

This novel approach lies in the area between offline and online ensemble approaches and can be seen either as a restriction of the former or an enhancement of the latter.

We identify some basic strategies that can be used to populate this reservoir and present our main contribution, dubbed Greedy Edge Expectation Maximization (GEEM), that maintains the reservoir content in the case of Boosting by viewing the samples through their projections into the weak classifier response space. We propose an efficient algorithmic implementation which makes it tractable in practice, and demonstrate its efficiency experimentally on several compute-vision data-sets, on which it outperforms both online and offline methods in a memory constrained setting.

## 1   Introduction

Learning a boosted classifier from a set of samples $S = \{X, Y\}^N \in \mathbb{R}^D \times \{-1, 1\}$ is usually addressed in the context of two main frameworks. In offline Boosting settings [10] it is assumed that the learner has full access to the entire dataset $S$ at any given time. At each iteration $t$, the learning algorithm calculates a weight $w_i$ for each sample $i$ – the derivative of the loss with respect to the classifier response on that sample – and feeds these weights together with the entire dataset to a weak learning algorithm, which learns a predictor $h^t$. The coefficient $a_t$ of the chosen weak learner $h^t$ is then calculated based on its weighted error. There are many variations of this basic model, too many to mention here, but a common aspect of these is that they do not explicitly address the issue of limited resources. It is assumed that the dataset can be efficiently processed in its entirety at each iteration. In practice however, memory and computational limitations may make such learning approaches prohibitive or at least inefficient.

A common approach used in practice to deal with such limitations is that of sub-sampling the data-set using strategies based on the sample weights $W$ [9, 13]. Though these methods address the limits of the weak learning algorithms resources, they nonetheless assume a) access to the entire data-set at all times, b) the ability to calculate the weights $W$ of the $N$ samples and to sub-sample $K$ of these, all in an efficient manner. The issues with such an approach can be seen in tasks such as computer vision, where samples need not only be loaded sequentially into memory if they do not all fit which in itself may be computationally prohibitive, but furthermore once loaded they must be pre-processed, for example by extracting descriptors, making the calculation of the weights themselves a computationally expensive process.

For large datasets, in order to address such issues, the framework of online learning is frequently employed. Online Boosting algorithms [15] typically assume access solely to a *Filter()* function, by which they mine samples from the data-set typically one at a time. Due to the their online nature

such approaches typically treat the weak learning algorithm as a black box, assuming that it can be trained in an online manner, and concentrate on different approaches to calculating the weak learner coefficients [15, 4]. A notable exception is the works of [11] and [14], where weak learner selectors are introduced, one for each weak learner in the ensemble, which are capable of picking a weak learner from a predetermined pool. All these approaches however are similar in the fact that they are forced to predetermine the number of weak learners in the boosted strong classifier.

We propose here a middle ground between these two extremes in which the boosted classifier can store some of the already processed samples in a *reservoir*, possibly keeping them through multiple rounds of training. As in online learning we assume access only to a *Filter()* through which we can sample $Q_t$ samples at each Boosting iteration. This setting is related to the framework proposed in [2] for dealing with large data-sets, the method proposed there however uses the filter to obtain a sample and stochastically accepts or rejects the sample based on its weight. The drawback of this approach is a) that after each iteration all old samples are discarded, and b) the algorithm must process an increasing number of samples at each iteration as the weights become increasingly smaller. We propose to acquire a fixed number of samples at each iteration and to add these to a persistent reservoir, discarding only a subset. The only other work we know which trains a Boosting classifier in a similar manner is [12], where the authors are solely concerned with learning in the presence of concept drift and do not propose a strategy for filling this reservoir. Rather they use a simple sliding window approach and concentrate on the removal and adding of weak learners to tackle this drift.

A related concept to the work presented here is that of learning on a budget [6], where, as in the online learning setting, samples are presented one at a time to the learner, a perceptron, which builds a classification model by retaining an active subset of these samples. The main concern in this context is the complexity of the model itself and its effect via the Gramm matrix computation on both training and test time. Subsequent works on budget perceptrons has led to tighter budgets [16] (at higher computational costs), while [3] proved that such approaches are mistake-bound.

Similar work on Support Vector Machines [1] proposed *LaSVM*, a SVM solver which was shown to converge to the SVM QP solution by adopting a scheme composed of two alternating steps, which consider respectively the expansion and contraction of the support vector set via the SMO algorithm. SVM budgeted learning was also considered in [8] via an $L_1$-SVM formulation which allowed users to specifically set a budget parameter $B$, and subsequently minimized the loss on the $B$ worst-classified examples.

As noted, these approaches are concerned with the complexity of the classification model, that is the budget refers to the number of samples which have none-zero coefficients in the dual representation of the classifier. In this respect our work is only loosely related to what is often referred to as budget learning, in that we solve a qualitatively different task, namely addressing the complexity of the parsing and processing the data during training.

Table 1: Notation

| | |
|---|---|
| $R_t$ | the contents of the reservoir at iteration $t$ |
| $|R_t|$ | the size of the reservoir |
| $Q_t$ | the fresh batch of samples at iteration $t$ |
| $\Sigma_{AA}$ | the covariance matrix of the edges $h \circ y$ |
| $\mu_A$ | the expectation of the edges of samples in set $A$ |
| $y_A$ | the vector of labels $\{-1, 1\}^{|A|}$ of samples in $A$ |
| $w^t$ | the vector of Boosting weights at iteration $t$ |
| $F_t(x)$ | the constructed strong classifier at iteration $t$ |
| $Filter()$ | a filter returning samples from $S$ |
| $h^t$ | the weak learner chosen at iteration $t$ |
| $\mathcal{H}$ | the family of weak learners |
| $\circ$ | component-wise (Hadamard) product |
| $T$ | number of weak learners in the strong classifier |

Table 2: Boosting with a Reservoir

Construct $R_0$ and $Q_0$ with $r$ and $q$ calls to $Filter()$.
**for** $t = 1, \ldots, T$ **do**
    Discard $q$ samples from $R_{t-1} \cup Q_{t-1}$ samples to obtain $R_t$
    Select $h^t$ using the samples in $R_t$
    Compute $a^t$ using $R_t$
    Construct $Q_t$ with $q$ calls to $Filter()$
**end for**
Return $F_T = \sum_{t=1}^{T} a_t h^t$

## 2 Reservoir of samples

In this section we present in more detailed form the framework of learning a boosted classifier with the help of a reservoir. As mentioned, the batch version of Boosting consists of iteratively selecting a weak learner $h^t$ at each iteration $t$, based on the loss reduction they induce on the full training set $S$. In the reservoir setting, weak learners are selected solely from the information provided by the samples contained in the reservoir $R_t$.

Let $N$ be the number of training samples, and $S = \{1, \ldots, N\}$ the set of their indexes. We consider here one iteration of a Boosting procedure, where each sample is weighted according to its contribution to the overall loss. Let $y \in \{-1, 1\}^N$ be the sample labels, and $\mathcal{H} \subset \{-1, 1\}^N$ the set of weak-learners, each identified with its vector of responses over the samples. Let $\omega \in \mathbb{R}_+^N$ be the sample weights at that Boosting iteration.

For any subset of sample indexes $B \subset \{1, \ldots, N\}$ let $y_B \in \{-1, 1\}^{|B|}$ be the "extracted" vector. We define similarly $\omega_B$, and for any weak learner $h \in \mathcal{H}$ let $h_B \in \{-1, 1\}^{|B|}$ stands for the vector of the $|B|$ responses over the samples in $B$.

At each iteration $t$, the learning algorithm is presented with a batch of fresh samples $Q_t \subset S$, $|Q_t| = q$, and must choose $r$ samples from the full set of samples $R_t \cup Q_t$ at its disposal, in order to build $R_{t+1}$ with $|R_{t+1}| = r$, which it subsequently uses for training.

Using the samples from $R_t$, the learner chooses a weak learner $h^t \in \mathcal{H}$ to maximize $\langle h_{R_t}^t \circ y_{R_t}, w_{R_t}^t \rangle$, where $\circ$ stands for the Hadamard component-wise vector product. Maximizing this latter quantity corresponds to minimizing the weighted error estimated on the samples currently in $R_t$. The weight $a^t$ of the selected weak learner can also be estimated with $R_t$.

The learner then receives a fresh batch of samples $Q_{t+1}$ and the process continues iteratively. See algorithm in Table 2. In the following we will address the issue of which strategy to employ to discard the $q$ samples at each time step $t$. To our knowledge, no previous work has been published in this or a similar framework.

## 3 Reservoir Strategies

In the following we present a number of strategies for populating the reservoir, i.e. for choosing which $q$ samples from $R_t \cup Q_t$ to discard. We begin by identifying three basic and rather straightforward approaches. **Max Weights (Max)** At each iteration $t$ the weight vector $w_{R_t \cup Q_t}^t$ is computed for the $r + q$ samples and the $r$ samples with the largest weights are kept. **Weighted Sampling (WSam)** As above $w_{R_t \cup Q_t}^t$ is computed, then normalized to 1, and used as a distribution to sample $r$ samples to keep without replacement. **Random Sampling (Rand)** The reservoir is constructed by sampling uniformly $r$ samples from the $r + q$ available, without replacement.

These will serve mainly as benchmark baselines against which we will compare our proposed method, presented below, which is more sophisticated and, as we show empirically, more efficient. These baselines are presented to highlight that a more sophisticated reservoir strategy is needed to ensure competitive performance, rather than to serve as examples of state-of-the-art baselines.

Our objective will be to populate the reservoir with samples that will allow for an optimal selection of weak learners, as close as possible to the choice we would make if we could keep all samples.

The issue at hand is similar to that of feature selection: The selected samples should be *jointly informative* for choosing the good weak learners. This forces to find a proper balance between the individual importance of the kept samples (i.e. choosing those with large weights) and maximizing the heterogeneity of the weak learners responses on them.

## 3.1 Greedy Edge Expectation Maximization

In that reservoir setting, it makes sense that given a set of samples $A$ from which we must discard samples and retain only a subset $B$, what we would like is to retain a training set that is as representative as possible of the entire set $A$. Ideally, we would like $B$ to be such that if we pick the optimal weak-learner according to the samples it contains

$$h^* = \underset{h \in \mathcal{H}}{\operatorname{argmax}} \langle h_B \circ y_B, w_B \rangle \tag{1}$$

it maximizes the same quantity estimated on all the samples in $A$. i.e. we want $\langle h_A^* \circ y_A, w_A \rangle$ to be large.

There may be many weak-learners in $\mathcal{H}$ that have the exact same responses as $h^*$ on the samples in $B$, and since we consider a situation where we will not have access to the samples from $A \setminus B$ anymore, we model the choice among these weak-learners as a random choice. In which case, a good $h^*$ is one maximizing

$$E_{H \sim \mathcal{U}(\mathcal{H})} \left( \langle H_A \circ y_A, \omega_A \rangle \mid H_B = h_B^* \right), \tag{2}$$

that is the average of the scores on the full set $A$ of the weak-learners which coincide with $h^*$ on the retained set $B$.

We propose to model the distribution $\mathcal{U}(\mathcal{H})$ with a normal law. If $H$ is picked uniformly in $\mathcal{H}$, under a reasonable assumption of symmetry, we propose

$$H \circ y \sim \mathcal{N}(\mu, \Sigma) \tag{3}$$

where $\mu$ is the vector of dimension $N$ of the expectations of weak learner edges, and $\Sigma$ is a covariance matrix of size $N \times N$. Under this model, if $\bar{B} = A \setminus B$, and with $\Sigma_{A,B}$ denoting an extracted sub-matrix, we have

$$E_{H \sim \mathcal{U}(\mathcal{H})} \left( \langle H_A \circ y_A, \omega_A \rangle \mid H_B = h_B^* \right) \tag{4}$$

$$= E_{H \circ y \sim \mathcal{N}(\mu, \Sigma)} \left( \langle H_A \circ y_A, \omega_A \rangle \mid H_B = h_B^* \right) \tag{5}$$

$$= \langle h_B^* \circ y_B, \omega_B \rangle + E_{H \circ y \sim \mathcal{N}(\mu, \Sigma)} \left( \langle H_{\bar{B}} \circ y_{\bar{B}}, \omega_{\bar{B}} \rangle \mid H_B = h_B^* \right) \tag{6}$$

$$= \langle (h_B^* \circ y_B), w_B \rangle + \langle \mu_{\bar{B}} + \Sigma_{\bar{B}B} \Sigma_{BB}^{-1} (h_B^* \circ y_B - \mu_B), w_{\bar{B}} \rangle \tag{7}$$

Though the modeling of the discrete variables $H \circ y$ by a continuous distribution may seem awkward, we point out two important aspects. Firstly the parametric modeling allows for an analytical expression for the calculation of (2). Given that we seek to maximize this value over the possible subsets $B$ of $A$, an analytic approach is necessary for the algorithm to retain tractability. Secondly, for a given vector of edges $h_B^* \circ y_B$ in $B$, the vector $\mu_{\bar{B}} + \Sigma_{\bar{B}B} \Sigma_{BB}^{-1} (h_B^* \circ y_B - \mu_B)$ is not only the conditional expectation of $h_{\bar{B}}^* \circ y_{\bar{B}}$, but also its optimal linear predictor in a least squares error sense.

We note that choosing $B$ based on (7) requires estimates of three quantities: the expected weak-learner edges $\mu_A$, the co-variance matrix $\Sigma_{AA}$, and the weak learner $h^*$ trained on $B$. Given these quantities, we must also develop a tractable optimization scheme to find the $B$ maximizing it.

## 3.2 Computing $\Sigma$ and $\mu$

As mentioned, the proposed method requires in particular an estimate of the vector of expected edges $\mu_A$ of the samples in $A$, as well as the corresponding covariance matrix $\Sigma_{AA}$.

In practice, the estimation of the above depends on the nature of the weak learner family $H$. In the case of classification stumps, which we use in the experiments below, both these values can be calculated with small computational cost.

A classification stump is a simple classifier $h_{\theta, \alpha, d}$ which for a given threshold $\theta \in \mathbb{R}$, polarity $\alpha \in \{-1, 1\}$, and feature index $d \in \{1, \ldots, D\}$, has the following form:

$$\forall x \in \mathbb{R}^D, \ h_{\theta, \alpha, d}(x) = \begin{cases} 1 & \text{if } \alpha x^d \geq \alpha \theta \\ -1 & \text{otherwise} \end{cases} \tag{8}$$

where $x^d$ refers to the value of the $d_{th}$ component of $x$.

In practice when choosing the optimal stump for a given set of samples $A$, a learner would sort all the samples according to each of the $D$ dimensions, and for each dimension $d$ it would consider stumps with thresholds $\theta$ between two consecutive samples in that sorted list.

For this family of stumps $H$ and given that we shall consider both polarities, $E_h(h_A y_A) = 0$.

The covariance of the edge of two samples can also be calculated efficiently, with $O(|A|^2 D)$ complexity. For two given samples $i,j$ we have

$$\forall h \in H, y_i h_i y_j h_j \in \{-1, 1\}. \tag{9}$$

Having sorted the samples along a specific dimension $d$ we have that for $\alpha = 1$, $y_i h_i y_j h_j \neq y_i y_j$ for those weak learners which disagree on those samples i.e. with $\min(x_i^d, x_j^d) < \theta < \max(x_i^d, x_j^d)$. If $I_j^d, I_i^d$ are the indexes of the samples in the sorted list then there are $(|I_j^d - I_i^d|)$ such disagreeing weak learners for $\alpha = 1$ (plus the same quantity for $\alpha = -1$), given that for each dimension $d$ there correspond $2(|A| - 1)$ weak-learners in $H$, we reach the following update rule $\forall d, \forall \{i, j\}$ :

$$\Sigma_{AA}(i, j)+ = y_i y_j (2 * (|A| - 1) - 4 * |I_j^d - I_i^d|) \tag{10}$$

where $\Sigma_{AA}(i, j)$ refers to the $i, j$ element of $\Sigma$. As can be seen, this leads to a cost of $O(|A|^2 D)$. Given that commonly $D \gg |A|$, this cost should not be much higher than $O(D|A| \log |A|)$ the cost of sorting along the $D$ dimensions.

### 3.3   Choice of $h^*$

As stated, the estimation of $h^*$ for a given $B$ must be computationally efficient. We could further commit to the Gaussian assumption by defining $p(h^* = h), \forall h \in H$ i.e. the probability that a weak learner $h$ will be the chosen one given that it will be trained on $B$ and integrating over $H$, this however, though consistent with the Gaussian assumption, is computationally prohibitive. Rather, we present here two cheap alternatives both of which perform well in practice.

The first and simplest strategy is to use $\forall B, h^* \circ y_B = (1, \dots, 1)$ which is equivalent to making the assumption that the training process will results in a weak learner which performs perfectly on the training data $B$. This is exactly what the process will strive to achieve, however unlikely it may be.

The second is to generate a number $|H_{Lattice}|$ of weak learner edges by sampling on the $\{-1, 1\}^{|B|}$ lattice using the Gaussian $H \circ y \sim \mathcal{N}(\mu_B, \Sigma_{BB})$ restricted to this lattice and to keep the optimal $h^* = \operatorname{argmax} h \in H_{Lattice}\langle(h_B \circ y_B), w_B\rangle$. We can further simplify this process by considering the whole set $A$ and the lattice $\{-1, 1\}^{|A|}$ and simply extracting the values $h_B^*$ for the different subsets $B$. Though much more complex, this approach can be implemented extremely efficiently, experiments showed however that the simple rule of $\forall B, h^* \circ y_B = (1, \dots, 1)$ works just as well in practice and is considerably cheaper. In the following experiments we present results solely for this first rule.

### 3.4   Greedy Calculation of $\operatorname{argmax}_B$

Despite the analytical formulation offered by our Gaussian assumption, an exact maximization over all possible subsets remains computationally intractable. For these reason we propose a greedy approach to building the reservoir population which is computationally bounded.

We initialize the set $B = A$, i.e. initially we assume we are keeping all the samples, and calculate $\Sigma_{BB}^{-1}$. The greedy process then iteratively goes through the $|B|$ samples in $B$ and finds the sample $j$ such that for $B' = B \setminus \{j\}$ the value

$$\langle \Sigma_{\bar{B}'B'} \Sigma_{B'B'}^{-1}(h_{B'}^* \circ y_{B'}), w_{\bar{B}'}\rangle + \langle h_{B'}^* \circ y_{B'}, w_{B'}\rangle \tag{11}$$

is maximized, where, in this context, $h^*$ refers to the weak learner chosen by training on $B'$. This process is repeated $q$ times, i.e. until $|\bar{B}| = q$, discarding one sample at each iteration.

In the experiments presented here, we stop the greedy subset selection after these $q$ steps. However in practice the subset selection can continue by choosing pairs $k, j$ to swap between the two steps. In our experiments however we did not notice any gain from further optimization of the subset $B$.

## 3.5 Evaluation of $E(\langle h_A^*, w_A \rangle | B)$

Each step in the above greedy process requires going through all the samples $j$ in the current $B$ and calculating $E(\langle h_A^*, w_A \rangle | B')$ for $B' = B \setminus \{j\}$.

In order for our method to be computationally tractable we must be able to compute the above value with a limited computational cost. The naive approach of calculating the value from scratch for each $j$ would cost $O(|B'|^3 + |B'||B|)$. The main computational cost here is the first factor, incurred in calculating the inverse of the covariance matrix $\Sigma_{B'B'}$ which results from the matrix $\Sigma_{BB}$ by removing a single row and column. It is thus important to be able to perform this calculation with a low computational cost.

### 3.5.1 Updating $\Sigma_{B'B'}^{-1}$

For a given matrix $M$ and its inverse $M^{-1}$ we would like to efficiently calculate the inverse of $M_{-j}$ which is results from $M$ by the deletion of row and column $j$.

It can be shown that the inverse of the matrix $M_{e_j}$ which results from $M$ by the substitution of row and column $j$ by the basis vector $e_j$ is given by the following formula:

$$M_{e_j}^{-}1 = M^{-1} - \frac{1}{M_{ii}}M_{j*}^{-1}M_{*j}^{-1} + e_j^T e_j \tag{12}$$

where $M_{*j}$ stands for the vector of elements of the $j_{th}$ column of matrix $M$ and $M_{j*}$ stand for the vector of elements of its $j_{th}$ row. We omit the proof (a relatively straightforward manipulation of the Sherman-Morrison formulas) due to space constraints. The inverse $M_{-j}^{-1}$ can be recovered by simply removing the $j_{th}$ row and column of $M_{e_j}^{-1}$.

Based on this we can compute $\Sigma_{B'B'}^{-1}$ in $O(|B|^2)$. We further exploit the fact that the matrices $\Sigma_{\bar{B}'B'}$ and $\Sigma_{B'B'}^{-1}$ enter into the calculations through the products $\Sigma_{B'B'}^{-1} h_{B'}^*$ and $w_{\bar{B}}^T \Sigma_{\bar{B}'B'}$. Thus by pre-calculating the products $\Sigma_{BB}^{-1} h_B^*$ and $w_{\bar{B}}^T \Sigma_{\bar{B}B}$ once at the beginning of each greedy optimization step, we can incur a cost of $O(|B|)$ for each sample $j$ and an $O(|B|^2)$ cost overall.

## 3.6 Weights $\tilde{w}_B$

*GEEM* provides a method for selecting which samples to keep and which to discard. However in doing so it creates a biased sample $B$ of the set $A$, and consequently weights $w_B$ are not representative of the weight distribution $w_A$. It is thus necessary to alter the weights $w_B$ to obtain a new weight vector $\tilde{w}_B$ which will takes this bias into account. Based on the assumption (3) and (7), and the fact that $\mu_A = 0$, we set

$$\tilde{w}_B = w_B + w_{\bar{B}}^T \Sigma_{\bar{B}B} \Sigma_{BB}^{-1} \tag{13}$$

The resulting weight vector $\tilde{w}_B$ used to pick the weak-learner $h^*$ correctly reflects the entire set $A = R_t \cup Q_t$ (under the Gaussian assumption)

## 3.7 Overall Complexity

The proposed method *GEEM* comprises, at each boosting iteration, three main steps: (1) The calculation of $\Sigma_{AA}$, (2) The optimization of $B$, and (3) The training of the weak learner $h_t$

The third step is common to all the reservoir strategies presented here. In the case of classification stumps by presorting the samples along each dimension and exploiting the structure of the hypothesis space $H$, we can incur a cost of $O(D|B| \log |B|)$ where $D$ is the dimensionality of the input space.

The first step, as mentioned, incurs a cost of $O(|A|^2 D)$ if we go through all dimensions of the data. However the minimum objective of acquiring an invertible matrix $\Sigma_{AA}$ by only looking at $|A|$ dimensions and incurring a cost of $O(|A|^3)$. Finally the second step as analyzed in the previous section, incurs a cost of $O(q|A|^2)$.

Thus the overall complexity of the proposed method is $O(|A|^3 + D|A|log|A|)$ which in practice should not be significantly larger than $O(D|B|log|B|)$, the cost of the remaining reservoir strategies. We note that this analysis ignores the cost of processing incoming samples $Q_t$ which is also common to all strategies, dependent on the task this cost may handily dominate all others.

# 4 Experiments

In order to experimentally validate both the framework of reservoir boosting as well as the proposed method *GEEM*, we conducted experiments on four popular computer vision datasets.

In all our experiments we use logitboost for training. It attempts to minimize the logistic loss which is less aggressive than the exponential loss. Original experiments with the exponential loss in a reservoir setting showed it to be unstable and to lead to degraded performance for all the reservoir strategies presented here. In [14] the authors performed extensive comparison in an online setting and also found logitboost to yield the best results. We set the number of weak learners $T$ in the boosted classifier to be $T = 250$ common to all methods. In the case of the online boosting algorithms this translates to fixing the number of weak learners.

Finally, for the methods that use a reservoir – that is *GEEM* and the baselines outlined in 3 – we set $r = q$. Thus at every iteration, the reservoir is populated with $|R_t| = r$ samples and the algorithm receives a further $|Q_t| = r$ samples from the filter. The reservoir strategy is then used to discard $r$ of these samples to build $R_{t+1}$.

## 4.1 Data-sets

We used four standard datasets: **CIFAR-10** is a recognition dataset consisting of $32 \times 32$ images of 10 distinct classes depicting vehicles and animals. The training data consists of 5000 images of each class. We pre-process the data as in [5] using code provided by the authors. **MNIST** is a well-known optical digit recognition dataset comprising 60000 images of size $28 \times 28$ of digits from $0 - 9$. We do not preprocess the data in anyway, using the raw pixels as features. **INRIA** is a pedestrian detection dataset. The training set consists of 12180 images of size $64 \times 128$ of both pedestrians and background images from which we extract HoG features [7]. **STL-10** An image recognition dataset consisting of images of size $96 \times 96$ belonging to 10 classes, each represented by 500 images in the training set. We pre-process the data as for CIFAR.

## 4.2 Baselines

The baselines for the reservoir strategy have already been outlined in 3, and we also benchmarked three online Boosting algorithms: **Oza** [15], **Chen** [4], and **Bisc** [11]. The first two algorithms treat weak learners as a black-box but predefine their number. We initiate the weak learners of these approaches by running Logitboost offline using a subset of the training set as we found that randomly sampling the weak learners led to very poor performance; thus though they are online algorithms, nonetheless in the experiments presented here they are afforded an offline initialization step. Note that these approaches are not mutually exclusive with the proposed method, as the weak learners picked by *GEEM* can be combined with an online boosting algorithm optimizing their coefficients.

For the final method [11], we initiated the number of selectors to be $K = 250$ resulting in the same number of weak learners as the other methods. We also conducted experiments with [14] which is closely related to [11], however as it performed consistently worse than [11], we do not show those results here.

Finally we compared our method against two sub-sampling methods that have access to the full dataset and subsample $r$ samples using a weighted sampling routine. At each iteration, these methods compute the boosting weights of all the samples in the dataset and use weighted sampling to obtain a subset $R_t$. The first method is a simple weighted sampling method (**WSS**) while the second is Madaboost (**Mada**) which combines weighted sampling with weight adjustment for the sub-sampled samples. We furthermore show comparison with a fixed reservoir baseline (**Fix**), this baseline subsamples the dataset once prior to learning and then trains the ensemble using offline Adaboost, the contents of the reservoir in this case do not change from iteration to iteration.

# 5 Results and Discussion

Table 3, 4, and 5, list respectively the performance of the reservoir baselines, the online Boosting techniques, and the sub-sampling methods. Each table also presents the performance of our GEEM approach in the same settings.

|  | **Max** | | **Rand** | | **WSam** | | **GEEM** | |
|---|---|---|---|---|---|---|---|---|
| Dataset | r=100 | r=250 | r=100 | r=250 | r=100 | r=250 | r=100 | r=250 |
| CIFAR | 29.59 (0.59) | 29.16 (0.71) | 46.02 (0.35) | 45.88 (0.24) | 48.92 (0.34) | 50.09 (0.24) | 50.96 (0.36) | 54.87 (0.28) |
| STL | 30.20 (0.75) | 30.72 (0.82) | 39.25 (0.32) | 39.40 (0.25) | 41.60 (0.39) | 42.93 (0.30) | 42.40 (0.65) | 45.70 (0.38) |
| INRIA | 95.57 (0.49) | 96.31 (0.37) | 91.54 (0.49) | 91.72 (0.35) | 94.29 (0.23) | 94.63 (0.30) | 97.21 (0.21) | 97.52 (0.13) |
| MNIST | 66.74 (1.45) | 68.25 (0.81) | 79.97 (0.24) | 79.59 (0.22) | 83.96 (0.29) | 84.07 (0.23) | 84.66 (0.30) | 84.33 (0.33) |

Table 3: Test Accuracy on the four datasets for the different reservoir strategies

|  | Online Boosting | | | **GEEM** |
|---|---|---|---|---|
| Dataset | **Chen** | **Bisc** | **Oza** | (r=250) |
| CIFAR | 39.40 (1.91) | 45.03 (0.93) | 49.16 (0.40) | 54.87 (0.28) |
| STL | 33.09 (1.49) | 36.35 (0.49) | 39.98 (0.56) | 45.70 (0.38) |
| INRIA | 94.23 (0.97) | 95.65 (0.38) | 95.50 (0.49) | 97.53 (0.13) |
| MNIST | 80.99 (1.11) | 85.25 (0.82) | 84.85 (0.54) | 84.33 (0.33) |

Table 4: Comparison of GEEM with online boosting algorithms

|  | **WSS** | | **Mada** | | **Fix** | | **GEEM** | |
|---|---|---|---|---|---|---|---|---|
| Dataset | r=100 | r=250 | r=100 | r=250 | r=1,000 | r=2,500 | r=100 | r=250 |
| CIFAR | 50.38 (0.38) | 51.66 (0.30) | 48.87 (0.26) | 49.44 (0.33) | 48.41 (0.88) | 52.40 (0.77) | 50.96(0.36) | 54.87 (0.28) |
| STL | 42.54 (0.35) | 44.07 (0.31) | 41.36 (0.32) | 42.34 (0.24) | 42.04 (0.19) | 46.07 (0.41) | 42.40 (0.65) | 45.70 (0.38) |
| INRIA | 94.24 (0.30) | 94.65 (0.16) | 94.26 (0.27) | 94.65 (0.10) | 92.46 (0.67) | 93.82 (0.74) | 97.21 (0.21) | 97.53 (0.13) |
| MNIST | 84.21 (0.27) | 84.51 (0.16) | 79.00 (0.33) | 78.99 (0.31) | 85.37 (0.33) | 88.02 (0.15) | 84.66 (0.30) | 84.33 (0.33) |

Table 5: Comparison of GEEM with subsampling algorithms

As can be seen, *GEEM* outperforms the other reservoir strategies on three of the four datasets and performs on par with the best on the fourth (MNIST). It also outperforms the on-line Boosting techniques on three data-sets and on par with the best baselines on MNIST. Finally, GEEM performs better than all the sub-sampling algorithms. Note that the **Fix** baseline was provided with ten times the number of samples to reach a similar level of performance.

These results demonstrate that both the reservoir framework we propose for Boosting, and the specific GEEM algorithm, provide performance greater or on par with existing state-of-the-art methods. When compared with other reservoir strategies, *GEEM* suffers from larger complexity which translates to a longer training time. For the INRIA dataset and $r = 100$ *GEEM* requires circa 70 seconds for training as opposed to 50 for the **WSam** strategy, while for $r = 250$ *GEEM* takes approximately 320 seconds to train compared to 70 for **WSam**. We note however that even when equating training time, which translates to using $r = 100$ for *GEEM* and $r = 250$ for **WSam**, *GEEM* still outperforms the simpler reservoir strategies. The timing results on the other 3 datasets were similar in this respect.

Many points can still be improved. In our ongoing research we are investigating different approaches to modeling the process of evaluating $h^*$, of particular importance is of course that it is both reasonable and fast to compute, one approach is to consider the maximum a posteriori value of $h^*$ by drawing on elements in extreme value theory.

We have further plans to adapt this framework, and the proposed method, to a series of other settings. It could be applied in the context of parallel processing, where a dataset can be split among CPUs each training a classifier on a different portion of the data.

Finally, we are also investigating the method's suitability for active learning tasks and dataset creation. We note that the proposed method GEEM is not given information concerning the labels of the samples, but simply the expectation and covariance matrix of the edges.

**Acknowledgments**

This work was supported by the European Community's Seventh Framework Programme FP7 - Challenge 2 - Cognitive Systems, Interaction, Robotics - under grant agreement No 247022 - MASH.

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
