[Reviews · NeurIPS 2013]

Submitted by Assigned_Reviewer_2

This paper propose a boosting algorithm GEEM keeping a limited number of samples in a reservoir while incremental learning. The key of GEEM is the gaussian apprixmation of the distribution of match of hypothesis for each sample and this paper shows the effectiveness of this simple approximation. GEEM keep updating the approximation and choosing the optimal hypothesis based on the approximation and selecting the samples to reserve with greedy subset selection. In the experiments againsts 3 reservor strategies, 3 online boosts and 3 sub-sampling methods for typical datasets, it is shown that GEEM outperforms other methods for most of datasets. I think quality, clarity, originality and significance are good.
Summary: This paper proposes a pool-based boosting algorithm GEEM is based on the gaussian approximation of the match of hypothesis of each sample and shows the effectiveness of this approximation in the experimets against typical online and subsampling methods. I think quality, clarity, originality and significance are good.

Submitted by Assigned_Reviewer_5

Summary of the paper:

The authors describe a boosting technique in which each weak learner is trained
on a small subset of the training sample as in FilterBoost and related
techniques. In each iteration, the subset is doubled and half of the total set
is then discarded. The deletion is based on maximizing the expected edge of the
classifier on the full reservoir set, given a Gaussian model of the pointwise
classifications.

Detailed remarks (in order of appearance):

26: computer

36: learns

86: 23?

96: What is A here?

214: It's strange to put an index in the exponent, especially d, seems like an
exponentiation.

232: It's strange to use this pseudo-code-like notation instead of "static"
math. Why not just simply define Sigma_AA as a sum over d of the right hand
side?

295: enter

330: samples don't have an edge.

331: The instability using the exponential loss is a major concern. Where does
it come from? Why is it stable for the logistic loss? Can you show any results
on whether the algorithm converges using any loss?

The experiments are meaningful in the sense that they show that the proposed
sampling strategy is better then the other strategies, given the same number of
iterations and pool size. Still, what is the practical goal? We do subsampling
for saving on computational time or memory, so I would like to see the errors
not after a given number of iterations, but spending the same computational
time. The overhead is asymptotically not that big, but still, the small
difference between WSam and GEEM might be compensated by the fact that WSam can
use more samples or can iterate a bit more while GEEM using its overhead for
smarter sampling.

Summary: The idea is interesting, the paper is well-written, the computational
complexity, which is a crucial aspect of the algorithm, is well analyzed. The
experiments show a slight but significant improvement over the simpler WSam
strategy, although I would like to see the results not at the same number of
iterations, but at the same computational (training) budget. My main theoretical
concern is that no boosting-like convergence theorems are shown.

Submitted by Assigned_Reviewer_6

This paper proposes a new boosting method that represents a tradeoff between online and offline learning. The main idea of the method is to maintain a reservoir of training examples (of fixed size) from which to train the weak learners. At each boosting iteration, new examples are added to the reservoir and then a selection strategy is used to reduce the reservoir to its original fixed size before the weak learner is trained. Several naive selection strategies are proposed but the main contribution of the paper is a more sophisticated selection strategy whose goal is to remove examples from the reservoir so that a weak learner trained on the reduced set will minimize the error computed on the whole set before reduction. The resulting algorithm is applied on four computer vision datasets, where it is shown to outperform several other online boosting methods.

The idea of using a reservoir is original and very interesting. The derivation of the GEEM algorithm from this initial idea is technically sound and the authors have been very careful in designing an efficient solution. Experiments are done on only four datasets but these datasets represent real computer vision problems of practical interest. GEEM is compared with many different methods, including some well-chosen baselines for the reservoir framework. The authors seem to have taken great care to put other methods in the best conditions for the comparison and despite this, the performance of GEEM is clearly superior to all other methods.

While I think that the reservoir idea is very interesting per se, it's not clear which particular learning setting or constraints the authors want to address with this idea in the paper. Do they want to reduce computing times, to reduce memory usage, to obtain the best possible predictive accuracy, or a combination of those? This lack of a clear definition of the problem makes the assessment of the significance of the method difficult. It clearly works better than other online methods in terms of accuracy, but computing times and memory usage seem worse. In terms of predictive performance, GEEM is also probably worse than offline boosting (but this baseline is actually not provided). I think that the paper should more clearly define the specific constraints the method is targeting and then make the experiments better reflect such constraints. At the moment, only test accuracies are compared in the experiments and the authors have only put a constraint on the number T of weak classifiers. In such conditions, offline boosting is probably the best method.

Some design choices also are not totally convincing. In Section 3.1, the idea of selecting the subset B so that it is as representative as possible of the entire set A does not seem so natural to me. Ideally, the subset B should be selected so that the weak learner that will be obtained from it will improve as much as possible the current model. Making it as close as possible to the weak learner trained from A does not make sense when A is not a good subset to train a weak learner. So, a more natural selection criterion from B should not necessarily depend on A. The choice of a perfect model h* in 3.3 is justified but why not instead simply look for the optimal weak learner from A and then project it on all Bs when doing the greedy search. The computational cost of this operation would probably be negligible with respect to the cost of the computation of the covariance matrix.

The paper is well written and mostly clear but it would be very difficult however to re-implement the algorithm given its complexity and also to reproduce the experiments as the settings of all tested algorithms are mostly unknown (see some questions below). The authors will however share their implementation upon publication.

Other comments
- I find it difficult to see how logitboost could be combined with the algorithm in Table 2, since logitboost as defined in (Friedman, Hastie, and Tibshirani, 1998) uses weak learners in the form of regressors and updates both the weights and the responses of these examples. Giving a description of the whole algorithm could help.
- It's not clear how comparable are the different algorithms in the experiments. In particular, are all algorithms actually seeing the exact same number of examples, ie. is the number of calls to the filter function the same in each case?
- In the algorithm of Table 2, why is h^t built from R_t and not from R_{t-1} U Q_{t-1}? Since R_t is selected so that the weak classifier built from it will look as much as possible like the weak classifier built from R_{t-1} U Q_{t-1} and since the computational cost for deriving R_t is higher than the computational cost for selecting h^t from R_{t-1} U Q_{t-1}, I don't see any reason not to train h^t from R_{t-1} U Q_{t-1}.
- The authors show that the FIX method needs 10 times more examples at each iteration than GEEM to reach similar accuracy. However, FIX is forced to always use the same examples at each iteration while GEEM samples 100 or 250 new examples at each iteration, meaning that at the end, GEEM has seen 25,000 or 62,500 fresh examples in total. So, I'm not sure the comparison is really fair. I would be interested to see the results of the application of offline boosting on the same number of examples as seen by GEEM, to see what we loose by working with a fixed reservoir.
- Also, why is FIX using adaboost while GEEM uses logitboost?
- The size r seems to be a sensible parameter that should not be chosen too small (as the weak learner is trained on r examples). The algorithm can thus not be turned into a real online method. I would like to see a study of the impact of the value of this parameter, in particular to see how large r has to be to reach a reasonable accuracy.
Summary: A technically sound and original boosting algorithm representing an interesting tradeoff between offline and online learning, with good performance on four computer vision datasets. The problem setting or constraints targeted by this algorithm are however not clear.
Author Feedback

Author rebuttal: We would like to thank the reviewers for their helpful comments and take the opportunity to address some of their concerns.

Assigned Reviewer 6 asks for some clarification concerning the motivation behind the work. Our goal is to investigate learning and specifically boosting in a setting of limited memory, i.e. in a setting where only a limited amount of data can be loaded in memory and used for training at each iteration. Thus we would like to investigate the area between online (no memory) and offline (no memory constraints) learning, showing that GEEM outperforms online methods (i.e. the use of a reservoir helps) as well as sampling techniques such as MadaBoost and WSS which can be used when memory is limited (though these methods typically have access to the entire dataset, something not afforded to the reservoir strategies).

To our knowledge there is no prior work in this field as previous work in budgeted learning seems to budget the classifier representation (kernel expansion) rather than the amount of working data allowed to be kept in memory.

Concerning the Fix baseline, we meant it to serve as an indication of the performance of offline boosting. We show that offline boosting needs 10 times more memory to obtain the same results as GEEM. If Fix were to see as many samples as GEEM (i.e. the entire dataset) it would need, for CIFAR, 200 times more memory than GEEM. We could add results for such a setting, as the reviewer suggested, to show the effect of memory constraints. Of the presented baselines, only Fix is restricted in the samples it views, the remaining baselines view the same number of samples. Finally we note that Fix uses Adaboost as in this case there is no instability and we obtained slightly better results than with the logistic loss.

Assigned Reviewer 5 raises concerns with regards to the instability of the exponential loss. With such an aggressive loss, the weight of misclassified
samples may be orders of magnitude higher than the weights of all properly classified ones. Hence, particularly in the case of noisy
labels, the choice of weak-learners oscillate, taking care of one population of samples, then another, etc. The obvious way of fixing
such behavior would be through the use of a learning rate, close in spirit to stochastic gradient descent. However, to keep the article's
message clear, we preferred to use at that point a less aggressive loss as a simpler way of regularizing the learning.

Assigned Reviewer 5 also mentions that he would like to see results with equalized budgets. We would like to point to the paragraph at lines 410-415 where we point out that even when equalizing computation time GEEM outperforms the other reservoir strategies. These results should probably be highlighted and expanded.

Assigned Reviewer 6 argues that perhaps h_t should be trained on R_{t-1} \cup Q_{t-1}. We could train h_t thus and then use GEEM to select R_t, we have run experiments in such a framework, the results showed that GEEM still outperforms the other reservoir strategies (and other baselines). We would like to note however that the cost of training on set A is not necessarily negligible compared to computing the covariance matrix \Sigma, as noted in section 3.7 the cost of computing \Sigma is O(|A|^3), depending on the dimensionality of the data D, this could be similar to O(D|A|log|A|), the cost of training on |A|.

Another concern raised by Assigned Reviewer 6 is that of the strategy for choosing B and h*. For the former, we aim to construct a set B such that training on B will yield a weak learner that performs well on the entire set A, given that at iteration t the data at our disposable is the set A, this seems to us a natural strategy, pick a training set B that is representative of the entire set A. For the latter we refer to our argument in the previous paragraph as well as to the fact that the current choice of h* has a cost of O(1).